



**The dynamics of spatio-temporal droughts in northeast brazil**

Joao Dehon Pontes Filho[1,2], Francisco Assis de Souza Filho[2], Ticiana Marinho de Carvalho Studart,[2]
Ályson Brayner Souza Estácio[1], Eduardo Sávio Passos Rodrigues Martins[1]

[1]Research Institute for Meteorology and Water Resources (FUNCEME), 60115-221 Fortaleza-CE, Brazil

[2]Hydraulic and Environmental Engineering Department (DEHA), Federal University of Ceará (UFC), 60020-181 Fortaleza-CE, Brazil.

C*orrespondence to*: Joao D. Pontes Filho[1] (joao.pontes@funceme.br)

**Abstract.** This study presents a new framework for simplifying the spatio-temporal dynamics of droughts.

The framework balances the complexity of drought analysis with the need for clear, actionable information. Using growth curves, growth rates, and acceleration of drought characteristics, the study provides a novel method to monitor drought severity, affected areas, and progression paths. By examining patterns and relationships between these characteristics, the framework identifies critical regions for targeted drought monitoring. The findings highlight that droughts originating in certain areas of Northeast

Brazil pose greater risks, requiring closer attention. This approach offers decision-makers a more effective tool to predict and mitigate drought impacts, improving drought management strategies.

## 1 Introduction

Droughts are a major global threat, impacting water resources, agriculture, and ecosystems

(Wilhite, 2000). It has a slow development and becomes noticeable when having serious impacts (Dracup et al., 1980). Climate change is expected to exacerbate drought impacts, particularly in vulnerable regions like Northeast Brazil, a semi-arid area with a long history of severe droughts (Marengo et al., 2017). Effective water resource management in drought-prone regions requires a thorough understanding of the spatio-temporal characteristics of droughts.

Spatio-temporal variability is a defining feature of droughts. Droughts can exhibit significant spatial heterogeneity, with some areas experiencing more severe impacts than others. Furthermore, droughts are dynamic, with their severity, area, and path changing over time (Andreadis et al., 2005;



Vicente-Serrano, 2006). Capturing these complexities is crucial for accurate drought assessment and prediction.

Despite their spatio-temporal characteristics, studies of droughts began by analyzing only the time component, based on the theory of runs (Yevjevich V, 1967). By using run theory, each drought event can be analyzed separately from the original time series and it gained popularity in drought analysis (Espinosa et al., 2019; Liu et al., 2019; Shiau, 2006). To start considering the spatial component, a basic approach to explicit this characteristic of droughts is analyzing the drought within a targeted and fixed

spatial extent. Studies following this approach usually regionalize the precipitation data within the targeted area using Thiessen polygons or homogenous regions using statistical clusterization such as Principal Component Analysis (PCA) or K-means (Portela et al., 2015; Vicente-Serrano, 2006; Zhou et al., 2019). However, these approaches are limited as droughts do not respect physical, political, or homogenous borders.

Currently, a new line of research on the characteristics of droughts is developing, characterized by the ability to understand how droughts move in time and space. Starting from the concept of severity-area-duration (SAD) curves, proposed by Andreadis et al.  (2005), followed by the development of the 3D (longitude, latitude, and time) clustering algorithm and other studies such as Herrera-Estrada et al. (2017) and Diaz et al (2019) who tracked how drought moves to understand drought

dynamics, demonstrating that 3D drought analysis is a promising improvement area for better drought monitoring and early warning.

These studies have advanced our understanding of how droughts evolve in time and space, but they have raised a new question, defined here as the dilemma of complexity. Reduce dimensionality, ignoring other faces of the problem, or over-complexify, making it difficult for managers and decision

makers to analyze results. For example, Wen et al. (2020) used a 3D clustering algorithm to analyze a large number of drought characteristics (drought duration, drought area, drought mass, drought volume, drought density, drought aggregation index, and longitude and latitude coordinates of centroid). Also, Herrera-Estrada and Diffenbaugh (2020) used seven different drought characteristics to characterize landfalling droughts from a 3D analysis. Many drought characteristics analyzed make it difficult for the

decision-maker to take early action due to the high amount of information. Furthermore, the



understanding of how characteristics accelerate and decelerate in time and space has not yet been analyzed. Understanding how each event evolves in terms of its spatio-temporal characteristics is important to improve early warning and to analyze mitigation actions.

To fill this complexity dilemma gap, we propose a simplified framework to analyze drought spatio-temporal characteristics comprehensively. This framework first analyses how each drought event evolves according to three new drought characteristics proposed here to analyze drought patterns, growth curve, growth rate, and acceleration. Then, attempts to understand the mean characteristics of drought events according to the place where its centroid originated. The Northeast Brazil region was chosen as a case study as it is a drought-prone area that during 2012-2018 experienced the most extreme drought ever recorded (Pontes Filho et al., 2020). In recent years, Brazil has been advancing its official drought monitoring (Martins et al., 2015). However, little spatiotemporal analysis has been made and there is a need for more research on how to use spatiotemporal analysis to improve drought characterization and management in Northeast Brazil (Brito et al., 2021).

The difficulty in monitoring the drought onset and its evolution over time and space is an important obstacle to proactive drought management (Liu et al., 2018). This study will provide valuable information for drought monitoring, forecasting, and management and contribute to the advancement of how we can mitigate drought's negative impacts.

## 2 Materials and methods

### 2.1 Data

This study was performed in Northeast Brazil, which is known as a drought-prone area that faces recurrent multi-year droughts. We used monthly total precipitation data from the University of East Anglia/Climate Research Unit (CRU), CRU TS v 4.05 at 0.5° x 0.5° resolution from 1950 – 2018 (Harris et al., 2020). The CRU TS v 4.05-time series has been available since 1901, but we chose only to use data from the mid-20th century onwards as the region did not have many rain gauges at the beginning of the century and the dataset interpolation may add noise for the grid analysis.



## 2.2 Drought definition

The spatio-temporal drought definition is divided into four steps: (i) data processing, (ii) drought definition in one dimension (1D), (iii) drought definition in two dimensions (2D), and (iv) drought definition in three dimensions (3D).

The first step for drought spatio-temporal definition is data processing is the first step. It requires gridded data to support the spatial analysis. Meteorological drought is often classified as below-average precipitation for a given area. As precipitation is the most worldwide climatic information, we chose this variable to make further analysis. However, the methodology proposed in this study for the drought spatio-temporal analysis fits any grided time-series of drought index and any drought definition

(e.g. hydrological

The second step is to define drought. The well-known Standardized Precipitation Index (SPI) (Mckee et al., 1993) was used because it is the drought index recommended by the World Meteorological Organization. The SPI has three main advantages: (i) its simplicity as it uses only precipitation; (ii) presents standardized information, making it easy to compare its values across different regions; and (iii)

can provide information in different timescales, being useful for agricultural, hydrological, and socioeconomic drought analysis using different time scales (Hayes et al., 2007; Pontes Filho et al., 2019).

The perception of drought varies according to the specific interests of users, making it impractical to have a single fully adequate definition (Palmer, 1965). Users perceive droughts differently because the water shortage can affect them at different times. Shorter time scales, such as 1 to 3 months,

can be more critical to agricultural users who do not irrigate their cultures. Longer time scales, such as 6 to 12 months, may relate to hydrological impacts on urban and irrigation water supplies. Thus, the aggregation period and the threshold selected can strongly impact drought analysis. We analyzed SPI for the 3, 6, 12, and 24 time-scale.

The identification of spatial-temporal drought is the third step. It involves dividing

interconnected grids into clusters and extracting continuous grids at both spatial and temporal scales to construct a three-dimensional (longitude, latitude, and time) drought structure. The algorithm uses the following sequence.





The gridded precipitation data is converted to SPI values using the selected time-scale. Run theory is applied for each grid cell as the traditional 1D analysis. It identifies the periods when the drought index is below the drought threshold. The values are converted into a binary format (0/1), assigning whether it is under drought conditions (1) or not (0).

For the 2D analysis, the algorithm scans all grid cells below the drought threshold for each time or snapshot. When the first cell value equals "1", indicating that the grid is under drought, the algorithm creates a drought event and searches for the 9x9-1 neighbors, excluding the center grid. If a neighbor cell also has the "1" value, it is included in the same drought event, forming a cluster. A new cluster event is created if another cell in the same snapshot has the "1" value, but it is not contiguous to any previous cluster. A minimum initial area of 1.6% of the total is required to analyze more regional events (Xu et al., 2015).

The 3D analysis connects spatial clusters through time. We link clusters with overlapping grid cells between time $t$ and time $t + 1$. An ID is assigned to each spatial cluster. If two or more different clusters are merged sometime after their formation, the ID of the oldest cluster (i.e. the lower ID) is conserved for the new merged cluster (Fig. 1). Moreover, the ID of previously disjoint clusters are updated to the ID of the merged one, since they are a snapshot of a single drought event evolving in space and time. In $t + 1$, a minimum overlap area is required to eliminate ambiguous drought events. A threshold of overlap area of 1.6% was chosen in this study following (Li et al., 2020). Through this methodology, each drought event evolving in space and time is assigned to a unique ID.

A sensitive analysis of the temporal and spatial parameters used here, the time-scales used in the SPI, and the minimum initial area and minimum overlap area, are provided to understand how these parameters can affect drought characterization.

An important consideration in the proposed algorithm is that when a cluster splits into two or more clusters, they all retain the same initial ID. This is a modification of the algorithm used by Diaz et al. (2019) and Herrera-Estrada et al. (2017), whose analysis only preserved the areas of the largest clusters. We chose this path because droughts can occur simultaneously in different regions, due to different precipitation mechanisms affecting each region. Therefore, conserving only the largest area may





artificially interrupt an event that is still occurring, or even completely ignore an event that occurred

simultaneously but in different regions.

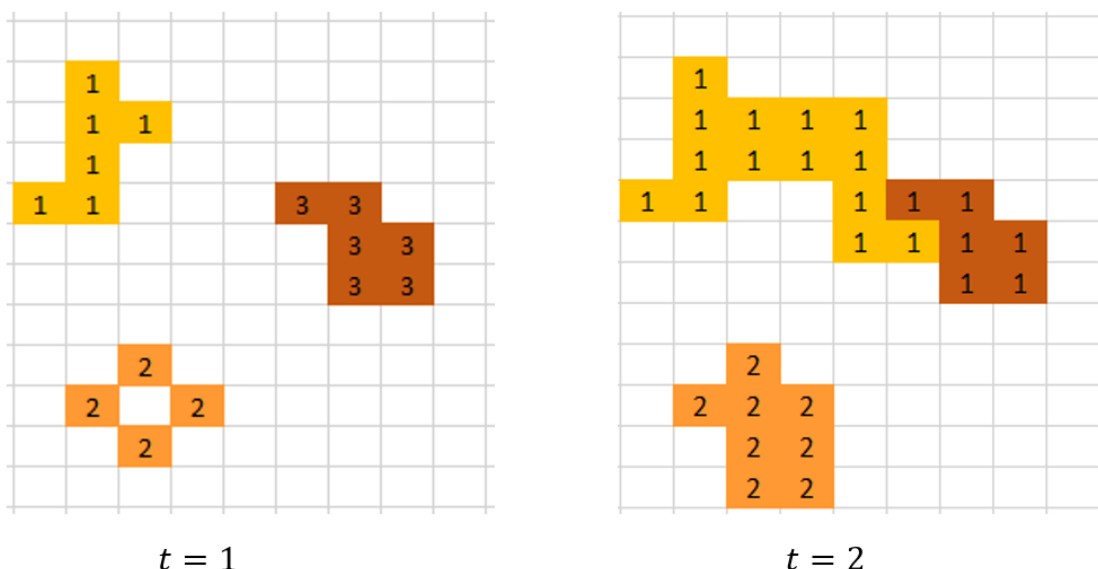

**Figure 1: Definition of spatial-temporal drought events by the three-dimensional clustering algorithm. Panel (a) shows three cluster**
**events at time t=1. Panel (b) shows two clusters at t=2.**

**2.2 Drought analysis**

After the clusters are defined, we divide the analysis of drought characteristics into two parts.

Firstly, we make an intra-event analysis of drought dynamics to understand drought dynamics during each

event. Then, we search for patterns and relationships between mean drought characteristics.

For the first part, three drought characteristics were studied: centroid, severity, and area.

Centroid is the geographic coordinates of the mass center of the affected area. The severity shows the sum

of severity at all gridded cells affected. The spatial extent informs how widespread the event was. Drought

duration was not chosen to be analyzed since it is present in all the other three drought characteristics.

To understand how the drought event evolves we propose to analyze droughts using three

evolution measures: Growth curve, growth rate, and acceleration analysis. The growth curve is the

cumulative sum of the characteristic value. The growth rate is the first derivative and shows the



instantaneous value at a given moment. Acceleration is the second derivative and informs whether the characteristic is intensifying or not (Table 1). The first application of growth curve, growth rate, and
acceleration analysis was proposed by Utsunomiya et al [(2020)] as an effort to monitor COVID-19 spread and is here adapted for understanding drought behavior.

**Table 1: Description of measures used in this study to characterize drought.**

| Characteristic | Evolution Measure | Equation | Description |
|---|---|---|---|
| **Centroid** | Growth Curve | $$d_t = \sum_{i=1}^{t} v_i$$ | Cumulative distance of the centroid path (km). Where $d_t$ is the centroid growth curve at instant $t$, and $v$ is the instantaneous centroid velocity. |
| | Growth Rate | $$v_t = dist(C_t, C_{t-1})$$ | Instantaneous centroid velocity (km/month). Where $v_t$ is the velocity at moment $t$, $C_t$ is the centroid position in the instant $t$, and $dist$ is a function that represents distance, in this case we used the Euclidian distance. |
| | Acceleration | $$a_{c,t} = \frac{(v_t - v_{t-1})}{dt}$$ | The rate of change of the centroid velocity with time (km/month²). Where $a_{c,t}$ is the instantaneous centroid acceleration at instant $t$, and $v$ is the instantaneous centroid velocity. |
| **Severity** | Growth Curve | $$M_t = \sum_{i=1}^{t} s_i$$ | The cumulative sum of the drought index value at the drought cluster ([SPI]). It gives the idea of the total magnitude of drought severity $s$. |
| | Growth Rate | $$s_t = \frac{\sum_{i=1}^{u} SPI_{i,t}}{u}$$ | Mean of cluster's drought index value at instant $t$ ([SPI]/month). The $SPI_{i,t}$ is the |



| | | | index value at grid cell $i$ and instant $t$; $u$ is the number of grid cells belonging to the cluster. |
|---|---|---|---|
| | Acceleration | $a_{s,t} = s_t - s_{t-1}$ | The rate of change of the severity with time ([SPI]/month2). Where $a_{s,t}$ is the instantaneous severity acceleration at instant $t$, and $s$ is the instantaneous severity. |

| Characteristic | Evolution Measure | Equation | Description |
|---|---|---|---|
| | Growth Curve | $CumA = \sum_{i=1}^{t} A_i$ | The cumulative sum of the drought area $A$ over the time the event persists (km²). |
| **Area** | Growth Rate | $A = \sum_{i=1}^{u} Area_{i,t}$ | The drought area at a given moment (km²/month). The $Area_{i,t}$ is the area of each grid cell $i$ and instant $t$. |
| | Acceleration | $a_{a,t} = A_t - A_{t-1}$ | The rate of change of the drought spatial extent with time (km²/month²). Where $a_{a,t}$ is the instantaneous area acceleration at instant $t$, and $A$ is the instantaneous area. |


The second part aims to understand drought patterns and the relationship between characteristics. To achieve this understanding, drought centroid, severity, area, and duration were analyzed. The centroid of onset and offset of drought events was used to investigate the possibility of having an area with a higher probability of starting more hazardous drought events. By dangerous, we mean events that are longer lasting, more severe, and affect widespread areas.





## 3 Results

### 3.1 Intra-event analysis of drought dynamics

The drought spatio-temporal analysis of droughts permits understanding where its onset was, how it evolved, the regions where it was affected, and where it remained the longest. This information is important to decision-makers to create coping measures to deal with future droughts. The 2012-2014 drought event in Northeast Brazil, for instance, is presented in Fig. 2 using SPI 12 and threshold -1.

The event started in 2012 in the south-west part of the studied region. During this year, drought expanded in area, affecting almost all the grid cells. By the middle of 2013, the drought reduced its impacted area and the severity. From the centroid path point of view, the drought started by traveling a great distance, but when it reached its maximum extent, its centroid remained stationary in the center of the study area. When the drought lost strength, the centroid remained in this area, as the drought was extinguished from the sides, remaining in the northern, central, and southern portions. The most affected area was the Central and South-West regions, with lower severity in the West. Figure 2 gives an overview of the drought event. It is possible to see the main affected areas and understand how drought evolved in time and space.

Other important drought events for understanding the spatio-temporal evolution of droughts in the study region are presented as supplementary material. The five main events selected were the 1950-1956, 1957-1961, 1980-1984, 2012-2014, and 2015-2018 droughts. These events were selected for their temporal scope, all with more than two years duration, and spatial, and the severity of the events and were also highlighted in other studies in the area (Hastenrath and Heller, 1977; Marengo et al., 2017; Pontes Filho et al., 2020). The 1950s proved to be a very dry period, with virtually the entire decade featuring drought in some regions. In general, all five events analyzed presented a predominance of the centroid by the central region of the study area.

Although the information brought by Fig. 2 is innovative and can give an overview of the drought, the discovery of patterns that can help plan and anticipate new events is only possible when comparing historical events. To do this, the strategy of analyzing average characteristics of events separated by the region of drought initiation was used and will be presented in the next section.



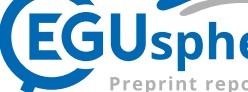

**Figure 2: Spatio-temporal analysis of the 2012-2014 drought event in Northeast Brazil using SPI 12, threshold -1. In (a), The gray cells are the ones that do not face drought, while the red cells are the ones under drought. When all the regions do not face drought, the region is marked in green. In (b), the red mark is where the drought centroid started, and the blue mark is where it finished. The black line tracks the centroid's path. In (c), (d), and (e), the red lines inform the magnitude of the growth curve, growth rate, and acceleration evolution measures.**





**3.2 Search for patterns and relationships between mean drought characteristics**

The analysis of drought mean characteristics was started by establishing the drought's onset region. The study area was divided into seven different regions and the mean characteristic of each event was analyzed.

The study area was divided into seven zones to try to understand if there are regions that are more prone to more dangerous events (Fig. 3). The seven zones were divided to better understand drought characteristics in regions affected by different climatic conditions. The northern part of the study area is mainly influenced by the Intertropical Convergence Zone (ITCZ), the eastern part is more influenced by the Southeast Trade Winds, and the southern part is mainly influenced by cold fronts. (Costa et al., 2018; Hastenrath, 2012; Nobre and Shukla, 1996; Uvo et al., 1998). Due to this divergence, it is important to understand the dynamics of droughts whose onset was influenced by these precipitation mechanisms.

Figure 3 presents a 4D analysis, by showing how drought characteristics using SPI3 related to each of the four dimensions (duration, severity, area, and region of centroid's onset). The severity and duration are in the y and x axes, the size of the bubble is related to the affected area and the color is related to where the region of centroid during its onset. It is clear from this picture that droughts from region 5 present the most important events in terms of duration, severity, and area. The characteristics of droughts from other areas do not present a clear pattern such as the ones from region 5. These results suggest that drought monitoring at region 5 is fundamental and higher-level observation should take place for droughts that are onset at this region as they have the potential to have more impacts in the region.

Despite the higher average values presented for droughts that were onset at region 5, drought characteristics for this region presented also higher variability. This result may be related to the fact that this area presents strong orographic influence, especially in the Southern part of this region, which tends to present a different behavior and increase noise in the classification of drought characteristics.

Upon analyzing the origin and path of drought centroids throughout the drought event, it was observed that droughts originating in the eastern part of the Northeast did not spread to other regions. However, events that originated further inland or in the northern part of the Northeast were able to develop and occupy other parts of the study area.





**Figure *3*: Boxplots of drought duration, area (km²), and severity for droughts in each region of Northeast Brazil using SPI 3. *The figure* also shows a 4D (duration, severity, area, and region of centroid's onset) analysis. The seven zones selected to analyze drought characteristics.**






The correlations between variables are presented in Fig. 4. Severity and area present a stronger correlation than each characteristic with duration, especially for the lower values. Region 5 shows most drought events, and the events with higher magnitude in every characteristic, showing that this region is more prone to develop droughts and to develop droughts with higher capacity to impact socio-economic 235 aspects in Northeast Brazil. Therefore, this region should be always well monitored by drought monitor.

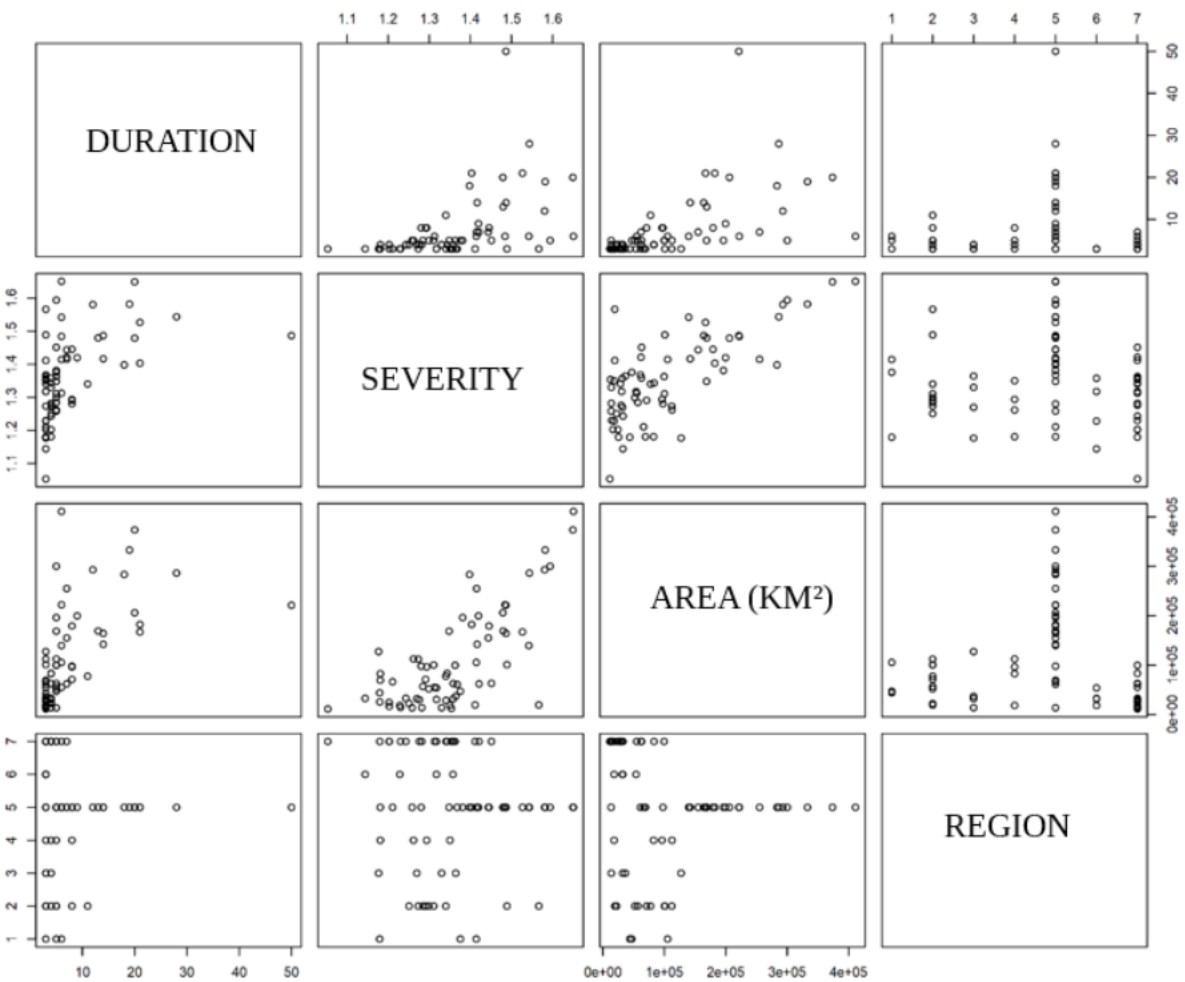

**Figure 4: Scatterplot of duration, severity area, and region of centroid's onset using SPI 3.**



### 3.3 Sensitivity analysis


A sensitivity analysis was carried out to investigate the temporal and spatial characteristics of drought. The temporal component was evaluated using different time scales of the Standardized Precipitation Index (SPI) at 3, 6, 12, and 24 months. For the spatial analysis, two parameters were tested, namely, the minimum initial area and the minimum overlap area.


Figure 5 presents the results of the temporal sensitivity analysis, which demonstrate that drought characteristics differ across various SPI time scales. Specifically, drought duration and centroid velocity increase with longer time scales, such as SPI 12 and SPI 24, which are associated with prolonged drought events and smaller centroid velocities. Longer time scales lead to less abrupt changes in the time series, making it more difficult for small precipitation events to end a prolonged drought. Additionally, longer aggregated periods preserve more information, resulting in smaller changes in the area between consecutive months. However, the signal for mean area and mean severity was not as clear. The SPI 6 showed a different behavior than the other time scales, as it showed a stronger variance. The strong interannual variation in the study area can cause noise in the 6-month aggregated time scale, which may explain these findings.






**Figure 5: Boxplot of the temporal sensitivity analysis of drought characteristics by considering different temporal time scales (3, 6, 12, and 24 months).**

For the spatial sensitivity analysis, we initially used a threshold of 1.6% for both the minimum initial area and the minimum overlap area. We kept the 1.6% threshold for one parameter while changing the other for the sensitivity analysis. Later, we performed the inversion procedure to analyze the other parameter. We conducted a sensitivity analysis of different spatial thresholds (0.8%, 1.6%, 3.2%, 4.8%) for SPI 12 for the two spatial parameters. For the studied area, the percentages correspond to 1, 7, 14, and 21 contiguous pixels required to initiate a drought event (minimum initial area) and connect the event





with the following month (minimum overlap area). To determine if these thresholds are too restrictive or too permissive, we conducted a sensitivity analysis (refer to Table 2).

**Table 2: Spatial sensitive analysis of drought events captured at different spatial thresholds (0.8%, 1.6%, 3,2% and 4,8%) for the two spatial parameters, minimal area and overlap area.**

| Thresholds | Minimal Area | | | Overlap Area | | |
|---|---|---|---|---|---|---|
| | Nº events | Duration (month) | Area (km²) | Nº events | Duration (month) | Area (km²) |
| 0.8% | 55 | 8 | 77.55 | 22 | 13 | 1149.72 |
| 1.6% | 22 | 13 | 1149.72 | 22 | 13 | 1149.72 |
| 3.2% | 17 | 20 | 2185.02 | 22 | 13 | 1149.72 |
| 4.8% | 15 | 25 | 3487.91 | 22 | 13 | 1149.72 |

The study found that the minimum area parameter strongly influenced the number of drought events recorded during the period. Lower values allowed for the identification of fewer and less extensive events with shorter durations and smaller areas. For example, the less restrictive threshold of 0.8% identified 55 drought events, while the more restrictive threshold of 4.8% identified only 15 events. Conversely, the duration and area variables showed an opposite effect. The median event was shorter and 275 less spread when the thresholds were lower. However, the number of drought events registered did not vary with the overlap area parameter, with 22 events for all thresholds.

The spatial sensitivity analysis revealed that the two parameters' thresholds had an uneven impact on the characterization of drought events. The minimum area criterion has a significant impact, defining many more isolated events that offset quickly, causing less environmental damage. The area 280 overlap criterion did not result in any changes in the definition of drought events.

Therefore, the importance of carefully considering the threshold values of temporal and spatial parameters when using the SPI to characterize drought in similar areas was demonstrated by the temporal and spatial sensitivity analysis.





**4 Discussion**

The analysis of drought dynamics during an event was divided into three stages: growth curve, growth rate, and acceleration. Previous studies have evaluated the spatio-temporal migration of drought tracking (Diaz et al., 2020; Zhou et al., 2019), but did not focus on the evolution of drought characteristics during the event. In this study, we propose using growth curves, growth rates, and accelerations to analyze changes in severity, area, and centroid path during each event. The proposed curves provide insights into

how droughts evolve over time and space by indicating when the event is accelerating or decelerating in each drought characteristic. This is important for understanding the different aspects of drought evolution. The curves were originally proposed by Utsunomiya et al (2020) to monitor the spread of COVID-19. The theoretical model for the spread of COVID-19 was clear and provided a formulation that could be used for forecasting. However, the same results were not found when analyzing drought characteristics.

The complex climatic variability and small number of drought events make it difficult to identify patterns within events. Therefore, we needed to shift our analysis to examine the mean behavior of different drought characteristics.

       The northeastern region of Brazil is known for its long, severe, and widespread drought periods, which are a result of strong spatio-temporal climatic variability due to a complex meteorological

system that influences rainfall patterns in the region. Oceanic and atmospheric factors, such as the Sea Surface Temperature (SST), the Atlantic dipole, and the El Niño Southern Oscillation (ENSO), play important roles in the occurrence of droughts in the region. The presence of the Atlantic dipole, which is characterized by a difference in SST between the north and south tropical Atlantic ocean temperature, has been linked to drought in the northeastern region of Brazil, especially in the northern part as it plays an

important role in the seasonal displacement of the ITCZ reaching its southernmost position from March to May (Hastenrath, 2012; Uvo et al., 1998). Less precipitation in the region is associated with warmer SST in the Tropical North Atlantic (TNA) and strong Southeast trade winds (Nobre & Shukla, 1996). When analyzing the last two millennia of precipitation in Northeast Brazil, Utida et al. (2019) found consistent dry periods when the ITCZ was in a mean position northern than its climatology. In contrast to

the northern region, precipitation on the eastern coast is modulated by breeze circulation and easterly wave disturbance (Gomes et al., 2015). In addition to these Atlantic Ocean factors, the ENSO also plays



an important role in influencing the precipitation patterns in the northeast (Hastenrath, 2012). During an El Niño event, the SST in the central and eastern Pacific Ocean rises, causing a shift in the atmospheric circulation that results in reduced rainfall in the northeastern region of Brazil. The region is also affected

by multidecadal variability (Kayano and Andreoli, 2004), highlighting the complexity of understanding and forecasting drought behavior in the region.

The northeastern region of Brazil is known for its prolonged, severe, and widespread droughts. This is due to the strong spatio-temporal climate variability caused by a complex meteorological system that affects rainfall patterns in the region. Droughts in the region are driven by oceanic and atmospheric

factors such as sea surface temperature (SST), the Atlantic Dipole, and the El Niño Southern Oscillation (ENSO).  The Atlantic dipole, characterized by a difference in SST between the north and south tropical Atlantic Ocean, has been linked to drought in northeastern Brazil. This is particularly true in the northern part of the region, as the dipole plays an important role in the seasonal displacement of the Intertropical Convergence Zone (ITCZ), which reaches its southernmost position from March to May (Hastenrath,

2012; Uvo et al., 1998). Warmer SST in the Tropical North Atlantic (TNA) and strong Southeast trade winds are associated with less precipitation in the region (Nobre & Shukla, 1996). Utida et al. (2019) found consistent dry periods in Northeast Brazil when the ITCZ was positioned north of its climatology over the last two millennia. Compared to the northern region, precipitation on the eastern coast is influenced by breeze circulation and easterly wave disturbance (Gomes et al., 2015). The ENSO also has

a significant impact on precipitation patterns in the northeast (Hastenrath, 2012). During an El Niño event, the SST in the central and eastern Pacific Ocean increases, leading to a change in atmospheric circulation that causes decreased rainfall in northeastern Brazil. The region is also affected by multidecadal variability (Kayano and Andreoli, 2004), highlighting the complexity of understanding and forecasting drought behavior in the region.

Given that the precipitation mechanisms that govern the northern and eastern regions of northeast Brazil differ, it is plausible that drought events starting in each region exhibit distinct characteristics. This paper advances the characterization of drought dynamics by revealing that droughts originating in the east do not develop to impact the entire region. However, droughts that originate in the central part of the Northeast are more likely to develop and affect the entire study area. This is due to the



ITCZ having greater difficulty displacing in central regions, resulting in longer, more severe, and more widespread droughts that begin in this area.

As no previous spatio-temporal study using the 3D clustering algorithm in Northeast Brazil has been conducted, it is difficult to compare the results found in this study. However, Brito et al. (2021) analyzed the spatio-temporal behavior of drought events in Northeast Brazil using the official Brazilian

Drought Monitor as a data source. They found that from 2014 to 2019, over 75% of the Northeast region of Brazil (NEB) area experienced exceptional drought. However, the methodology used only considered the duration and area of events as individual characteristics and did not utilize an integration technique to comprehend their relationships, such as the 3D cluster algorithm. In contrast, Silva et al. (2019) discovered that the more centralized regions of Northeast Brazil were drought hotspots, which is

consistent with the presented findings.

Sensitivity analysis was conducted using SPI 3, 6, 12, and 24. The results showed that different time scales have an impact on drought characteristics, particularly in terms of duration and centroid velocity. Longer time scales were associated with prolonged drought events, which had smaller centroid velocities and less abrupt changes in the time series. This made it difficult for small precipitation

events to end prolonged drought events. Moreover, longer aggregated periods preserve more information, resulting in smaller changes in the area between consecutive months. The study also found that the central area is more prone to prolonged, severe, and widespread drought events. Additionally, droughts that start on the eastern coast usually do not migrate to other central or northern areas of Northeast Brazil. The analysis of spatial sensitivity revealed that the number of recorded drought events is affected by the

minimum area criterion, while the overlap area criterion does not affect the number of recorded drought events. In this study, a threshold of 1.6% of the studied area was chosen for both the minimum initial area and minimum overlap area criteria following Xu et al. (2015), which was deemed sufficient to characterize drought events that have the potential to cause environmental and societal impacts.

**4 Conclusion**

An approach to characterizing drought based on spatio-temporal structure has been shown to provide valuable information for decision-makers. The analysis of this study was divided into two parts:

analyzing drought dynamics within events and searching for patterns and relationships between mean drought characteristics.

We proposed a novel method for visualizing the evolution of drought characteristics, displaying the growth curve, growth rate, and acceleration of severity, area, and centroid path. This approach helps decision-makers understand event progression, identify the most affected regions, and determine if the drought is improving or worsening.

The second analysis focused on discovering patterns to aid in creating proactive drought plans by comparing historical events. We analyzed the average characteristics of events based on the region of drought initiation. The study found that droughts originating from specific regions tend to cause more prolonged, severe and widespread impacts, suggesting a need for prioritized monitoring in these areas.

A spatial sensitivity analysis revealed that the choice of thresholds affects the characterization of drought events in terms of time and space. Longer aggregated periods and higher minimum area criteria are associated with more prolonged and widespread droughts. These findings highlight the importance of selecting appropriate thresholds for temporal and spatial parameters when using SPI for drought characterization in similar regions. The results enhance our understanding of drought dynamics, assisting in improved monitoring and prediction, and informing the creation of effective proactive drought plans.

**Code and data availability**

The code and data used in this study is available upon request by emailing the corresponding author.

**Author contributions**

Conceptualization, methodology and validation, J.D.P.F. and F.d.A.S.F.; formal analysis, investigation, writing—original draft preparation, J.D.P.F., F.d.A.S.F. and E.S.P.R.M.; software, J.D.P.F and ABSE.; writing—review and editing, supervision: F.d.A.S.F., E.S.P.R.M. and T.M.d.C.S.

**Acknowledgements**

We would like to thank the Federal University of Ceará (UFC) for their support in conducting this research, as well as the Ceará Foundation for Meteorology and Water Resources (FUNCEME) for their institutional support. We also acknowledge





the financial support provided by CAPES (Coordination for the Improvement of Higher Education Personnel), which was instrumental in making this study possible.


**Competing interests**

The authors declare no conflict of interest.

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
