# Peer review of "The dynamics of spatio-temporal droughts in northeast brazil"

_EGUsphere, 2024_

## Referee Comment (RC1)

**Review of the manuscript:** *The dynamics of spatio-temporal droughts in northeast brazil*

Anonymous Reviewer #

Pontes Filho et al. (2024) present a new framework for analysing and communicating spatio-temporal drought dynamics. The authors use an interesting approach derived from tools to monitor pandemics (such as COVID-19) to represent drought dynamics.

**General comments**

LANGUAGE: The manuscript has several typos and poorly-structured sentences. I have highlighted a few on the attached document. Please review the text.

MATERIALS AND METHODS: Please present the study area in this section. Th method presentation lacks clarity and variables are not well describe. This section would greatly benefit from a diagram or flowchart.

RESULTS: Figures have major flaws, such as missing labels and wrong values in axis. Some analyses (4D?) are also not presented in the methods. Also concepts such as correlation are misused.

DISCUSSION: The authors do not provide a proper discussion of their results. Major part of discussion (including a duplicated paragraph)

NOVELTY: The authors argue that the methods of using growth curve and rate are new, but I fail to see the technical gain of using these terms over the classical magnitude (accumulated severity) and total affected area. Regarding centroid tracking and spatio-temporal analysis, that is also not a novelty as illustrated by some of the papers literature (Zhou et al., 2019; Diaz et al., 2020; and specially Wen et al., 2020).

DECISION-MAKERS: The authors mention that their method provides a "more effective tool to predict and mitigate drought impacts" but fail to demonstrate how.

It is my opinion that the manuscript has major technical flaws and was not ready for submission. **My recommendation is for *rejection*.** After major work on these flaws, I'd recommend resubmission of their manuscript to this publishing house (either HESS or NHESS), as it fits well the profile.

[revised manuscript text omitted]